# BMP3 Deficiency Accelerates Cartilage-to-Bone Transition in Ectopic Bone

**DOI:** 10.3390/biomedicines13102508

**Published:** 2025-10-15

**Authors:** Viktorija Rumenović, Natalia Ivanjko, Nataša Kovačić, Slobodan Vukičević, Igor Erjavec

**Affiliations:** 1Scientific Centre of Excellence for Reproductive and Regenerative Medicine, School of Medicine, University of Zagreb, 10000 Zagreb, Croatia; viktorija.rumenovic@mef.hr (V.R.); natalia.ivanjko@mef.hr (N.I.); natasa.kovacic@mef.hr (N.K.); vukicev@mef.hr (S.V.); 2Laboratory for Molecular Immunology, Croatian Institute for Brain Research, School of Medicine, University of Zagreb, 10000 Zagreb, Croatia

**Keywords:** bone morphogenetic protein 3, ectopic bone, ossification, osteoprogenitors

## Abstract

**Background:** Ectopic bone formation models provide useful insights into bone tissue formation and remodeling processes. The use of a subcutaneous site emphasizes the focus on cytokine signaling and cell migration and proliferation while minimizing the effect of mechanical loading and direct interaction with surrounding stem cells. **Methods:** To study the effect of BMP3 on bone formation and remodeling, Bmp3^-/-^ mice were subcutaneously implanted with an autologous blood coagulum device containing BMP6, and bone formation was examined at days 7 and 14 post-implantation. Bone marrow cell composition was assessed using FACS. Formation of ectopic bone was analyzed using micro-CT, immunohistochemistry, and RNAseq to obtain transcriptomic data. **Results:** Bone marrow from Bmp3^-/-^ mice showed reduced lymphoid-lineage subsets, expanded myeloid lineage, and altered proportions of several osteochondroprogenitor subsets. A limited amount of newly formed bone tissue was seen in the implants after 7 days, while ectopic bone was more evident after 14 days, with significantly more bone in the Bmp3^-/-^ mice compared to WT mice. Localization of Sox9 and Runx2 showed a more advanced stage of bone tissue remodeling in Bmp3^-/-^ mice. Transcriptomic analysis showed upregulation of approximately 1700 genes on day 7 and 190 genes on day 14. **Conclusions:** These results suggest that BMP3 regulates the composition of bone and cartilage progenitor populations in bone marrow and consequently bone formation by arresting the remodeling of cartilage to bone tissue. The lack of BMP3 in ectopic bone accelerates the transition from the cartilaginous template to proper bone tissue.

## 1. Introduction

Bone tissue formation and the ability to regenerate are unique features in human and animal physiology, as they are governed by cells that incorporate organic compounds with inorganic salts [1]. During morphogenesis, cascades of various pathways and patterns are established, which results in properly formed bone tissue. The majority of bone tissue is formed through endochondral ossification, a process during which cartilaginous template, or anlage, is gradually replaced by bone [2,3]. Also, the ability of bone tissue to regenerate after injury invokes the same morphogenesis pattern cascades, which restore bone tissue structural stability. Mechanical forces exerted to bone tissue are crucial for bone development and growth, as bone tissue responds to external stimuli [4]. Although bone tissue has been comprehensively studied, due to the complexity of cellular and molecular events occurring during bone formation and remodeling, there are still shortcomings in current knowledge regarding these processes [5].

By applying a reverse engineering approach and creating de novo bone at an ectopic site, the process of bone formation can be further elucidated. This approach enables the removal of variables influencing bone tissue, which would be difficult to exclude during physiological bone formation or regeneration processes.

In our research, to further elucidate the role of bone morphogenetic protein 3 (BMP3) in bone formation, we sought to employ an ectopic bone formation model in mice. In this model, the bone is formed outside of its usual place of origin, mostly connected to other bone tissue, in contrast to orthotopic bone, which is formed in the correct anatomical position. A subcategory of ectopic ossification called heterotopic ossification (HO) represents bone formation in soft tissue, such as muscle, subcutaneous or fibrous tissue [6]. Heterotopic ossification in soft tissues gives us a chance to study the bone formation process in greater detail, as it removes the variable of mechanical loading and its influence on bone formation and remodeling [7,8]. By reducing the effect of stress/strain forces, the focus is shifted to cytokines, cell migration, and proliferation together with intracellular events happening during bone formation.

Bone morphogenetic proteins (BMPs) have diverse roles in the organism: they regulate many developmental processes, facilitate bone and cartilage formation and homeostasis and also maintain numerous organ functions [9,10,11]. The BMP family is a part of the transforming growth factor β (TGF-β) superfamily which binds to BMP receptors, thus activating Ser/Thr kinase pathways with concurrent activation of the Smad1/5/8 complex [12,13]. The signaling pathways that BMPs orchestrate are following a simple three-component system comprising of a ligand, receptor, and signal transducer, but they are quite versatile in all tissue types [14]. A major function of BMPs in bone tissue is to serve as signals for mesenchymal stem cell migration, proliferation, and differentiation into chondrocytes and osteoblasts [15]. In recent years, BMPs have been combined with various types of carriers to promote bone formation and healing [16].

Although BMP3 is a part of the BMP protein family, it lacks the osteogenic potential found in other members of the BMP family. It acts through a different signaling pathway, binding to the activin receptor type 2b (Acvr2b), with its role considered antagonistic to other osteogenic BMPs [17,18]. BMP3 is expressed in bone tissue, lungs, kidney, and the intestine [19,20], and it is the most abundant BMP in demineralized bone [21,22]. Mice lacking BMP3 have an increased bone mass in an otherwise identical phenotype compared to wild type (WT) mice, confirming that this protein negatively regulates bone density [17]. On the other hand, mice with overexpressed BMP3 show delayed endochondral ossification with spontaneous rib fractures [23]. BMP3 seems to prevent uncontrolled bone formation by antagonizing osteogenic BMPs in bone tissue. It also regulates chondrocytes in the epiphyseal area since mice lacking BMP3 show increased trabecular bone formation with locally overexpressed Runx2 [20]. BMP3 overexpression in bone marrow-derived mesenchymal stromal cells (BMSCs), however, results in increased chondrogenic differentiation activity with increased expression of Sox9, as well as collagen type 2 [24].

In this study (Appendix A), we sought to characterize bone marrow composition in Bmp3^-/-^ mice, with an emphasis on osteochondroprogenitor subsets. We explored BMP3 regulation of ectopic bone formation in mice lacking functional BMP3, where we investigated early and late ectopic bone formed in the subcutaneous space using micro-CT and histological analysis and analyzed gene expression. We also investigated the effect of functional BMP3 absence in ectopic bone tissue on a transcriptional level using RNA sequencing.

## 2. Methods

### 2.1. Animals

Bmp3^-/-^ mice used in the experiments were created using VelociGene technology described previously [18]. Briefly, Bmp3^-/-^ C57BL/6NTac mice were generated via a LacZ knock-in into exon 1 of the Bmp3 gene, rendering it nonfunctional. Mice were bred in the animal facility of the School of Medicine, University of Zagreb, Croatia and kept in a controlled environment with ad libitum access to food and water. The Ethics Committee of the School of Medicine and National Ethics Committees approved all procedures.

WT and Bmp3^-/-^ male and female mice aged between 12 and 16 weeks were used in the experiments. Genotypes were determined from a tail tissue sample applying standard genotyping procedures [25] using the following primers: LacZ: 5′-TTTCCATGTTGCCACTCGC-3′, 3′-ACCGCACGATAGAGATTCGG-5′, Bmp3: 5′-GAAGTAGAGCGGTGCGACAGCA-3′, 3′-AAGGTCCCTACAGTGTACCGCCA-5′.

### 2.2. Flow Cytometry

Flow cytometry analysis was performed as described previously [26]. Briefly, femora and tibiae were isolated from 10 WT and 10 Bmp3^-/-^ mice, washed in ice-cold complete MEM α (Capricorn Scientific, Ebsdorfergrund, Germany), and transferred to a sterile hood. Bones were then washed twice in sterile 0.1 M PBS and transferred to fresh complete MEM α. Bone ends were trimmed, and bone marrow was flushed with cold 0.1 M PBS using a syringe with a 26 G needle until pale. Bone marrow cells were then collected, centrifuged for 5 min at 250× *g* at 4 °C, and supernatant was discarded. Red blood cells were removed via incubation in ammonium chloride lysing buffer for 5 min at room temperature (RT). The cell pellet was then washed in 0.1 M PBS, single-cell suspensions were prepared by passing the cells through a 100 μm cell strainer, and suspensions containing 2–5 × 10^6^ cells were transferred to FACS tubes. Nonspecific antibody binding was blocked by adding 0.5 μL/sample of anti-mouse CD16/CD32 (93; BioLegend, San Diego, CA, USA) and incubating for 5 min at RT. After blocking, cells were stained with 6-color panels and the following antibodies: anti-mouse CD45-allophycocyanin (APC; 30-F11; BioLegend, San Diego, CA, USA), CD31-APC (390; BioLegend, San Diego, CA, USA), TER119-APC (TER119; BioLegend, San Diego, CA, USA), CD51-biotin (RMV-7; BioLegend, San Diego, CA, USA), CD90.2-FITC (30-H12; eBioscience, Thermo Fisher Scientific, Waltham, MA, USA), CD200-PE (OX90; eBioscience, Thermo Fisher Scientific, Waltham, MA, USA), CD105-PECy7 (MJ7/18; eBioscience, Thermo Fisher Scientific, Waltham, MA, USA), Sca-1-FITC (D7; eBioscience, Thermo Fisher Scientific, Waltham, MA, USA), CD140b-PECy7 (APB5 eBioscience, Thermo Fisher Scientific, Waltham, MA, USA), CD44-APCCy7 (IM7; BioLegend, San Diego, CA, USA), B220-FITC (RA3-6B2; eBioscience, Thermo Fisher Scientific, Waltham, MA, USA), Ly6G/Ly6Cy7 (RB6-8C5; BioLegend, San Diego, CA, USA), CD11b-PE (M1/70; eBioscience, Thermo Fisher Scientific, Waltham, MA, USA), CD3-APC (145-2C11; BioLegend, San Diego, CA, USA), and F4/80-APCCy7 (BM8; BioLegend, San Diego, CA, USA). Dead cells were labeled with 7-amino-actinomycin D (7-AAD; BioLegend, San Diego, CA, USA). After incubation for 30 min at 4 °C, cells were washed with PBS and incubated with streptavidin-APCCy7 (BioLegend, San Diego, CA, USA) for 30 min at 4 °C, washed again, and resuspended in 0.1 M PBS containing 7-AAD 10 μg/mL. Fluorescent signals were acquired using Attune instrument (Life Technologies, Thermo Fisher Scientific, Carlsbad, CA, USA) and analyzed using FlowJo v.10.10.0. software (FlowJo, Ashland, OR, USA) (Appendix A). The number of acquired events was 1–2 × 10^6^ for mesenchymal panels and 10^5^ for hematopoietic lineage panels.

### 2.3. Subcutaneous Ectopic Bone Formation Assay

A total of 34 WT and Bmp3^-/-^ mice of both sexes and similar age were selected, and ectopic bone formation was induced by employing a well-established model of induction using a whole blood coagulum device (WBCD) mixed with BMP6 [27]. To prepare the implant, 100 µL of full blood was drawn from the tail vein, mixed with 5 µg of BMP6, and left for 60 min at room temperature to fully coagulate.

Formed blood coagulum was prepared for implantation. Mice were anesthetized using i.p. injection of ketamine and xylazine (ketamine 100 mg/kg body weight, xylazine 10 mg/kg body weight). After sedation, a 1 cm incision was made above the sternum. The coagulum (1 cm × 0.4 cm) was implanted subcutaneously in the right axillary region. The cut was sutured and animals were monitored for 7 or 14 days as an early and late time point in ectopic bone formation. Upon experiment termination, the implants were isolated and fixed in 4% formaldehyde for micro-CT and histological analysis or in TRIzol reagent (Invitrogen, Waltham, MA, USA) and stored at −80 °C for RNA isolation.

### 2.4. Micro-CT Analysis

For the time point of 7 days, 13 WT and 8 Bmp3^-/-^ mice were analyzed. For the time point of 14 days, 12 WT and 18 Bmp3^-/-^ mice were analyzed. Collected implants were fixed and scanned using Skyscan 1076 micro-CT (Bruker, Kontich, Belgium) prior to histological analysis. The following scanning parameters were used: 40 kV/250 µA resulting in 9 µm resolution, 198° scan, with a rotation step of 0.5°, frame averaging set at 2, and 0.025 mm titanium filter to reduce beam hardening. The scans were reconstructed using NRecon v.1.7.4.6. software (Bruker, Kontich, Belgium) with a GPU-accelerated algorithm. Bone parameter analysis was performed using CTAn v.1.20.8.0. (Bruker, Kontich, Belgium). The resulting bone parameters are presented as bone tissue volume (BV), bone volume fraction (BV/TV), and trabecular number (Tb.N). Visualization and 3D model generation was performed using CTVox v.3.3.1. software (Bruker, Kontich, Belgium).

### 2.5. Histology and Immunohistochemistry

Samples for histological analysis from the 7 or 14 days ectopic bone experiments were decalcified in EDTA for three weeks, and 5 µm paraffin slices were made. For qualitative histological analysis, hematoxylin and eosin (H&E) staining was used, while immunohistochemical (IHC) analyses were performed to localize specific markers.

Formalin-fixed paraffin-embedded (FFPE) slices were first deparaffinized in xylene and rehydrated in descending concentrations of ethanol, followed by incubation in 1× PBS. H&E staining was performed following a standard procedure. Briefly, slides were incubated in hematoxylin solution for 3 min and then washed under running tap water for 30 min. Then they were placed in eosin stain for 1 min and washed in distilled water. After dehydration, slices were mounted using Canada balsam and analyzed using Olympus BX53 light microscope (Olympus, Tokyo, Japan).

The Mouse and Rabbit-Specific HRP/DAB IHC Detection Kit—Micro-polymer (Abcam, Cambridge, UK) was used for IHC after heat induced antigen retrieval (HIER) in citrate buffer pH 6.0. Primary antibody (rabbit anti-Runx2, 1:1000, Abcam, Cambridge, UK; rabbit anti-Sox9, 1:2000, Abcam, Cambridge, UK) was incubated overnight at 4 °C in a moist chamber, and the appropriate secondary antibody from the kit was incubated for 1.5 h at RT. Diaminobenzidine (DAB) was used as a chromogen, and the nuclei were counterstained with hematoxylin. Slides were mounted using Immunohistomount (Sigma-Aldrich, Burlington, MA, USA) and acquired using an Olympus BX53 light microscope (Olympus, Tokyo, Japan) and BP24 camera (Olympus, Tokyo, Japan).

### 2.6. RNA Sequencing

Two ectopic bone samples per group were mechanically homogenized in 1 mL of TRIzol (Invitrogen, Waltham, MA, USA) by Ultra-Turrax T25 homogenizer (IKA, Staufen, Germany). Total RNA was extracted using Direct-zol RNA Miniprep (Zymo Research, Irvine, CA, USA) according to the manufacturer’s instructions. The quality of total RNA was assessed using an Agilent 2100 Bioanalyzer (Agilent Technologies, Santa Clara, CA, USA).

The RNA-seq service was provided by Zymo Research (Zymo Research, Irvine, CA, USA). For mRNA sequencing, a cDNA library was constructed with Zymo-Seq RiboFree Total RNA Library Prep Kit (R3000, Zymo Research, Irvine, CA, USA) according to the manufacturer’s instructions. Briefly, RNA was reverse-transcribed into cDNA, which was followed by ribosomal RNA depletion. Next, partial P7 adapter sequence was ligated at the 3′ end of cDNAs, followed by second strand synthesis and partial P5 adapter ligation to the 5′ end of the double-stranded DNAs. Lastly, libraries were amplified to incorporate full-length adapters under the following conditions: initial denaturation at 95 °C for 10 min; 10–16 cycles of denaturation at 95 °C for 30 s, annealing at 60 °C for 30 s, and extension at 72 °C for 60 s; and final extension at 72 °C for 7 min. Successful library construction was confirmed with Agilent’s D1000 ScreenTape Assay on TapeStation (Agilent Technologies, Santa Clara, CA, USA). RNA-Seq libraries were sequenced using an Illumina NovaSeq (Illumina, San Diego, CA, USA) to a sequencing depth of at least 30 million read pairs (150 bp paired-end sequencing) per sample.

Bioinformatics analysis was carried out by using RNAseq pipelines that were built using Nextflow [28]. Briefly, quality control of raw reads was carried out using FastQC v0.11.9. Adapter and low-quality sequences were trimmed from raw reads using Trim Galore! v0.6.6. Trimmed reads were aligned to the reference genome using STAR v2.6.1d [29]. BAM file filtering and indexing was carried out using SAMtools v1.9 [30]. RNAseq library quality control was implemented using RSeQC v4.0.0 and QualiMap v2.2.2 [31,32]. Duplicate reads were marked using Picard tools v2.23.9. Library complexity was estimated using Preseq v2.0.3 [33]. Duplication rate quality control was performed using dupRadar v1.18.0 [34]. Reads overlapping with exons were assigned to genes using featureCounts v2.0.1 [35]. Classification of rRNA genes/exons and their reads were based on annotations and RepeatMasker rRNA tracks from UCSC genome browser when applicable. Differential gene expression analysis was completed using DESeq2 v1.28.0 with concurrent usage of an adjusted *p*-value of 0.01 to increase the stringency of significant gene identification [36]. Functional enrichment analysis was achieved using g:Profiler Python API v1.0.0 [37]. Quality control and analysis results plots were visualized using MultiQC v1.9 [38]. The genome assembly used for the mice was GRCm39 (Ensembl, #240).

### 2.7. RT-qPCR

To validate the RNA-seq data, differentially expressed genes involved in cartilage and bone formation were selected for RT-qPCR analysis. Tissue samples from day 14 (*n* = 6 per genotype) stored in TRIzol™ reagent (Invitrogen, Waltham, MA, USA) were mechanically homogenized, and RNA was isolated as previously described [39]. After measuring the RNA concentration, it was reverse-transcribed into cDNA using the RT kit (Applied Biosystems, Waltham, MA, USA). cDNA was diluted 1:10 in 1× TE buffer and used for qPCR. A commercial kit was used following instructions for qPCR (Takara, Shiga, Japan) in Light Cycler (Roche, Basel, Switzerland) with an amplification temperature of 60 °C and 35 cycles. Each sample was run in duplicates with negative controls (instead of cDNA, the same volume of 1× TE buffer was added). The primers used for this reaction are listed in Table 1. The results were analyzed using the ΔΔC_T_ method [40] with GAPDH as an internal control.

### 2.8. ELISA

To assess bone remodeling status in mice during ectopic bone formation, serum from 6 WT and Bmp3^-/-^ mice was collected pre-implantation and on day 7 or 14 following implantation. Bone formation was quantified by measuring the concentration of osteocalcin using the Mouse Osteocalcin ELISA kit (Abcam, Cambridge, UK), while bone resorption was quantified by measuring the concentration of bone sialoprotein using the Mouse Bone Sialoprotein ELISA kit (Abcam, Cambridge, UK) according to the manufacturer’s instructions. Biotek EL808 (Biotek, Winooski, Vermont, Canada) microplate reader was used for absorbance quantification.

### 2.9. Statistical Analysis

The distribution of all of the obtained data was first assessed using the Kolmogorov–Smirnov test. Data following a Gaussian distribution were examined with either an unpaired *t*-test (for comparisons between two groups) or a one-way ANOVA followed by Tukey’s multiple comparison test (for three or more groups). For non-Gaussian distributions, comparisons between two groups were carried out using the Mann–Whitney U test, whereas the Kruskal–Wallis test with Dunn’s post hoc analysis was applied when more than two groups were involved. Results are presented as mean ± standard deviation for normally distributed data or as median with the minimum and maximum values for non-normally distributed data. Statistical significance is indicated by asterisks (* *p* ≤ 0.05, ** *p* ≤ 0.01, *** *p* ≤ 0.001), with values of *p* < 0.05 considered significant. All analyses were performed using GraphPad Prism v.8.4.3. software (GraphPad Software, San Diego, CA, USA).

## 3. Results

### 3.1. Flow Cytometry

Flow cytometry analysis was performed on live cells. Gating strategies delineated hematopoietic and mesenchymal cell lineages (Appendix A). Among hematopoietic cells, both B-(B220^+^) and T-lymphocytes (CD3^+^) were reduced, while myeloid lineage cells (CD11b^+^Ly6C/G^+^) were more abundant in the bone marrow of Bmp3^-/-^ mice (Figure 1A). Amongst non-hematopoietic cells, the Sca-1^+^ population was upregulated, while the CD105^+^ population was downregulated in Bmp3^-/-^ mice in comparison to WT mice (Figure 1B,C). Osteochondroprogenitor subpopulations were not significantly different between groups in separate experiments, but the earliest skeletal progenitors seem to comprise a greater portion of the CD200^+^ population in Bmp3^-/-^ mice then in WT mice, while populations corresponding to committed cartilage progenitors were less abundant (Figure 1D).

### 3.2. Ectopic Bone Micro-CT Analysis

Autologous blood coagulum containing 5 µg of BMP6 was implanted in mice of appropriate age, sex, and genotype (Figure 2A). After 7 or 14 days, the resulting ectopic bone was imaged ex vivo using micro-CT.

During the early stage, after 7 days, bone formation occurred within a limited capacity, as seen with micro-CT imaging (Figure 2B). Although the whole implant was considerably larger in size, the extent of newly formed bone tissue detected using micro-CT was comparatively limited. Quantified bone parameters confirmed differences in bone formation in Bmp3^-/-^ animals (Figure 2C). Even at an early stage, Bmp3^-/-^ mice showed a trend toward more bone volume (BV, BV/TV) with a similar number of trabeculae (Tb.N) compared to WT mice, but no statistical significance was reached.

In the late stage, by day 14, micro-CT imaging revealed a substantial increase in bone formation. Compared to the results obtained from the 7-day time point, bone tissue was formed around the whole implant, forming a thin cortical bone with extensive trabecular bone within the implant (Figure 2B). A significant difference in bone parameters was seen in Bmp3^-/-^ mice when compared to WT mice. Bone volume (BV, BV/TV) and trabecular number (Tb.N) were significantly increased in Bmp3^-/-^ mice (Figure 2C).

### 3.3. Bone Turnover Markers

To analyze bone remodeling during ectopic bone formation, mouse serum was collected and analyzed using the specific ELISA kits. Bone resorption was examined by measuring the level of bone sialoprotein (BSP), while bone formation was examined by measuring the level of osteocalcin (OCL) in the serum on days 7 and 14 following WBCD implantation (Figure 2D). Bone sialoprotein was higher on day 7, indicating higher resorption levels at this time point compared to day 14 in both groups. Although the concentration was higher in Bmp3^-/-^ mice, there was no statistically significant difference between WT and Bmp3^-/-^ mice. Osteocalcin concentration peaked on day 7 in WT mice and was reduced to baseline levels by day 14, while a stable concentration was maintained in Bmp3^-/-^ mice during the 14-day period.

### 3.4. Ectopic Bone Immunohistochemistry Analysis

Immunohistological staining corroborated the micro-CT results, revealing more advanced ossification in Bmp3^-/-^ mice implants compared to WT after both 7 and 14 days, as indicated by Runx2 localization (Figure 3). On day 7, in WT mice, Runx2 expression was restricted to the surface region, while in Bmp3^-/-^ mice, it was localized at the site of mineralized tissue (Figure 3A). A greater number of Sox9-positive cells was observed in the Bmp3^-/-^ samples, but only around the implant, while at the site of mineralized tissue, Sox9 expression in cells was low. By day 14, Runx2 expression in Bmp3^-/-^ implants extended throughout the entire implant slice, suggesting the widespread presence of mature osteoblasts actively depositing bone matrix. In contrast, Sox9 expression was low in WT and Bmp3^-/-^ samples at this time point. Runx2-positive osteoblasts were also present in WT samples, indicating that endochondral ossification was still ongoing in WT animals on day 14 (Figure 3B).

### 3.5. Gene Expression Analysis

Transcriptomic analysis was used to determine the relative abundance of numerous genes in newly formed ectopic bone of WT and Bmp3^-/-^ mice on day 7 and 14 post-implantation (Figure 4A). Using heatmaps, gene expression patterns were revealed when comparing WT and Bmp3^-/-^ mouse samples on day 7 and 14 (Figure 4B). Among the top upregulated genes were *Bglap*, *Bglap2*, and *Msx2* as well as *Oscar* and *Ngp*. Notably, *Mmrn1* and *Adgrf4* (*Gpr115*), a gene important in dentin matrix [41], were also found to be upregulated in the analyzed Bmp3^-/-^ tissue samples. Transcriptomic profiling on day 14 revealed an upregulation of granulocyte-associated genes in Bmp3^-/-^ implants, including *Ngp*, *Mpo*, *Mmp8*, and *Prg2* (Appendix A).

RNA-seq analysis showed that 1633 genes were upregulated in the ectopic bone 7 days post implantation, whereas 1648 genes were upregulated after 14 days in Bmp3^-/-^ mice compared to WT mice (Figure 4C,D). Only 188 and 194 genes were upregulated in WT and Bmp3^-/-^ mice, respectively, in ectopic bone after 14 days when compared to 7 days. When comparing ectopic bone in WT and Bmp3^-/-^ mice, on day 7, no significant change in gene expression was observed after using an adjusted *p*-value of 0.01. Comparison of ectopic bone on day 14 showed significant upregulation of 136 genes in WT mice, while only 45 genes were upregulated in Bmp3^-/-^ mice. The top 50 differentially expressed genes between Bmp3^-/-^ and WT samples are listed for 7 days (Appendix A) and 14 days (Appendix A) post-implantation in the axillary region. To further corroborate transcriptomic data, *Runx2*, *Sox9*, and *Aggrecan* relative gene expression, as markers of osteoblast and chondroblast differentiation, were measured using qPCR on day 14 (Figure 4E). Upregulation of *Runx2* and downregulation of *Aggrecan* were found in Bmp3^-/-^ samples, while *Sox9* expression showed no significant difference.

Gene ontology and pathway analysis showed that on day 7, although with low numbers of enriched genes, there was a trend in the involvement of regulation of lymphatogenesis and cytokine receptor interaction, while genes enriched on day 14 were involved in the detection of chemical stimuli and sensory perception and G-protein coupled receptor signaling pathway (Appendix A). When comparing ectopic bone in WT and Bmp3^-/-^ mice on day 14, a higher number of upregulated genes in WT mice were involved in immune system processes, stress response, hematopoiesis, and myeloid cell differentiation (Appendix A).

## 4. Discussion

The aim of this study was to assess the effect of BMP3 on ectopic bone formation and the transition from cartilage to bone tissue. Our results showed enhanced bone formation potential in Bmp3^-/-^ mice, marked by earlier skeletal progenitors, fewer committed cartilage progenitor cells in the bone marrow, and enhanced ectopic bone formation compared to WT mice.

Until now, a limited number of studies have explored the effects of BMP3 on bone tissue, despite it being the most abundant BMP in bone. It has been shown that BMP3 has an antagonistic effect on BMP2 and BMP4 [17,18], but no other interactions with other members of the BMP family were examined. In this study, we investigated for the first time the unknown functions of BMP3 using an in vivo model of bone formation.

One of the most valuable results from our study was the observation that systemic lack of BMP3 enhances ectopic bone formation in vivo. Although ectopic bone formation does not completely mimic bone repair at an orthotopic site, it still recapitulates developmental events and signaling pathway cascades occurring during embryonic bone formation. Our animal model of ectopic bone formation enables us to circumvent the effect of mechanical loading on bone and focus on cellular events and molecular mechanisms that are occurring during de novo bone formation. This is especially important when exploring the role of BMP3 in these events, as it has been shown that mechanical loading reduces BMP3 expression during cartilage and bone tissue formation [42]. In Bmp3^-/-^ mice, the gene is already downregulated, and we have shown that bone formation is enhanced even without mechanical loading in these mice. Without mechanical loading of the ectopic bone, the BMP3 gene should not be downregulated in WT animals, and in turn, this would show us an actual effect of BMP3 protein on bone formation.

Another aspect of the subcutaneous ectopic bone model is that this environment is devoid of potential chondoprogenitor and osteoprogenitor stem cells, as opposed to an intramuscular ectopic bone model [6]. In order to form bone at this site, the implant must be positioned in the proximity of blood vessels to enable migration of appropriate stem cells [43]. In the axillary region, there is an abundance of blood vessels, which in turn secure sufficient vascularization of the implant and stem cell migration. To enable bone formation in the subcutaneous space, bone progenitor cells must migrate to this area. Since the fascia surrounding the subcutaneous space lacks such progenitor cells, the bone marrow is the most probable source of these migrating cells [6,44]. By using a subcutaneous ectopic bone formation model, we effectively excluded muscle satellite cells as potential progenitors, which are typically involved in the intramuscular ectopic bone formation model [6]. This approach places greater emphasis on bone marrow-derived progenitor cells as the primary contributors to bone formation.

Therefore, we have assessed the composition of bone marrow within long bones and observed alterations in the composition of both hematopoietic and non-hematopoietic compartments containing osteochondroprogenitors at various stages of development. Amongst populations delineated by a single marker, the Sca-1^+^ population was significantly expanded. Sca-1 is expressed in a non-hematopoietic progenitor population capable of trilineage differentiation [45]. Mice deficient in Sca-1 exert an age-dependent decrease in the self-renewal activity of bone marrow mesenchymal progenitors and impaired differentiation and function of osteoblasts [46], so an increase in this progenitor pool mediated by a lack of BMP3 signaling might contribute to the enhanced osteogenesis observed in this study. Furthermore, the population of CD51^+^CD105^+^ non-hematopoietic cells was reduced in Bmp3^-/-^ mice. CD105 is, according to Chan et al., expressed in more committed stages of skeletal progenitor development (bone–cartilage–stromal progenitor (BCSP), committed bone (BP), and cartilage (CP) progenitors), while the earliest skeletal progenitors express only CD200 [47]. CD105^+^ cells, corresponding to committed cartilage progenitors, seem to occupy a smaller proportion of the CD200^+^ population, while the population of CD200 single-positive cells is slightly enlarged. In summary, BMP3 seems to negatively regulate the earliest progenitor pool and enhance its maturation.

Ectopic bone models that mostly use a potent osteogenic molecule largely depend on the carrier to limit and localize the action of the molecule [48]. BMP6 implants, particularly when combined with autologous blood coagulum as a scaffold, have shown superior results in accelerating bone healing, as opposed to other BMPs [49,50]. Even in human clinical trials, recombinant human BMP6 applied within autologous blood coagulum has been demonstrated to accelerate bone healing in patients undergoing high tibial osteotomy [51]. In our experiments where osteogenic BMP6 was delivered in a blood coagulum to Bmp3^-/-^ mice, we observed a significant increase in the amount and quality of ectopic bone formation compared to WT controls. This result aligns with previous studies that have suggested that BMP3 acts as an antagonist to BMP2 and BMP4, which are also known to promote osteogenic differentiation and bone formation [52,53]. Micro-CT imaging enabled the visualization and characterization of the ectopic bone formed in Bmp3^-/-^ and WT mice at an earlier and later time point of 7 and 14 days post implantation, respectively. Implants obtained 7 days post-implantation were comparable in size to the implants obtained after 14 days, but the extent of ossification was much lower. Analysis of these implants showed that, even on day 7, the mineralized tissue parameters in Bmp3^-/-^ mice showed an upward trend when compared to WT mice, but it did not reach significant levels. This trend turned significant after 14 days, as Bmp3^-/-^ mice had more mineralized tissue. Although the extent of ectopic bone formation is limited and a dramatic increase in bone remodeling markers, such as bone sialoprotein, BSP, or osteocalcin, OCL, is not expected, the trends of these biomarkers may provide insights into the rate of bone remodeling in WT and Bmp3^-/-^ mice. A bone resorption marker, BSP, showed an increase on day 7 and a return to basal levels on day 14. A bone formation marker, OCL, showed a stark increase in WT animals, while it remained constant in Bmp3^-/-^ mice. This would indicate that the rise of OCL in Bmp3^-/-^ mice happened prior to day 7, and we were not able to capture its peak. Normal bone formation through endochondral ossification is a process that requires cartilage-to-bone transition, where cartilaginous template is replaced by proper bone tissue. This process has not been fully elucidated, but research has shown that the *Runx1* gene is a key signaling molecule for chondrocyte-to-osteoblast lineage commitment [54], as indicated by the expression of *Sox9* and *Runx2* in the 14 day period. In Bmp3^-/-^ mice, this transition was accelerated, and the cartilaginous tissue was replaced by bone in a much shorter timeframe, as evidenced by the results shown 7 days following implantation, indicating that BMP3 has a significant role in this process.

Based on histological slides, on day 7, the majority of the implants contained remaining erythrocytes encapsulated in the layer of fibroblasts, while mineralized tissue was limited in both WT and Bmp3^-/-^ mice. On day 14, complete removal of erythrocytes and fibroblasts was observed together with chondroblast and osteoblast cells proliferating and maturing. Novel bone formation was nearly complete after 14 days, as indicated by tissue morphology and *Runx2* expression. *Runx2* and *Sox9* are integral to the process of ossification in vivo, with *Sox9* primarily expressed during early chondrogenesis and *Runx2* more prominent during the later stages of osteogenesis [55]. The role of *Sox9* is crucial during early chondrogenesis by promoting cartilage template maturation necessary for subsequent ossification [56]. Conversely, *Runx2* plays a pivotal role during later stages by facilitating osteoblast differentiation; however, it is not essential in the final stages of differentiation to osteocytes [57]. The balance between these transcription factors ensures proper temporal progression from cartilage to bone. *Sox9* was even shown to have an inhibitory effect on osteoblast and chondrocyte maturation via repression of *Runx2* in vitro, which is a mechanism for osteochondroprogenitor cell fate determination [55]. On the other hand, its downregulation, with the appropriate levels of *Runx2*, leads to direct ossification in vitro in human MSCs [58]. BMP2, BMP4, and BMP6 are known to induce ectopic bone formation in vivo [27,59,60], and the interaction of BMP3 with BMP2 and BMP4 was studied. BMP3 inhibits the osteogenic activity of BMP2 by 50% in the MC3T3-E1 cell line when applied in 10× higher concentration [17] and reduces Smad1/5/8 phosphorylation induced by BMP4 [18]. Thus, BMP3 deficiency in Bmp3^-/-^ mice resulted in more bone during the ectopic bone formation assay due to the lack of attenuation of BMP6 activity in the WBCD carrier used in our model.

RNA sequencing data revealed early activation of osteogenic and immune pathways in Bmp3^-/-^ implants compared to the WT on day 7. Although only a few of the genes were differentially expressed at this early time point, the main reason for this could be the extent of the mineralized tissue formed on day 7. Since the implant mostly consisted of erythrocytes, which have no nucleus, and fibroblasts, the isolated mRNA from mineralized tissue was suppressed; thus, information regarding gene expression is lacking. The effect of BMP3 downregulation on fibroblasts has not been previously investigated; however, the present findings suggest that BMP3 does not influence fibroblast proliferation. One potential approach to address the limited RNA yield could involve excising the mineralized tissue and pooling the samples to obtain an adequate amount of mRNA, but this would introduce other complications such as sample contamination, proper delineation of mineralized tissue, and sample pooling variability. We encountered no limitations in implants obtained after 14 days, as they consisted mostly of mineralized bone and cartilaginous tissue, and the list of differentially expressed genes was extensive. At this time point, key upregulated genes in Bmp3^-/-^ indicate accelerated osteoblast maturation [61], the recruitment of osteoclast precursors [62], as well as early extracellular matrix remodeling and potential vascular involvement [41,63]. Transcriptomic profiling on day 14 showed an extensive upregulation of granulocyte-associated genes in Bmp3^-/-^ implants, indicating robust myeloid cell infiltration and activation and microenvironment shaping [64,65]. Elevated *Mmp8* expression also reflects active extracellular matrix remodeling, possibly facilitating accelerated bone turnover and maturation observed histologically [66]. These data align with changes in bone marrow composition observed in Bmp3^-/-^ mice, where the absence of BMP3 may influence immune–skeletal interactions, contributing to accelerated ossification.

qPCR results corroborate the previous findings. Implant tissue analysis confirmed an advanced remodeling stage in Bmp3^-/-^ samples on day 14, as seen by a 2-fold upregulation in *Runx2* expression. As mentioned previously, *Runx2* is a key transcription factor involved in osteoblast differentiation and bone formation. Its expression pattern suggests that BMP3 deficiency accelerates osteogenic commitment and bone matrix maturation [67]. Furthermore, significant downregulation of *Aggrecan*, a major cartilage matrix component, was also observed, indicating a transition from the cartilaginous template toward bone formation [68]. At this time point, we observed no change in *Sox9* gene expression between WT and Bmp3^-/-^ mice. These molecular changes are consistent with the histologically observed greater formation of mineralized tissue and further support a regulatory role for BMP3 in governing the timing and extent of skeletal tissue remodeling.

These findings point out that BMP3 regulates endochondral ossification and the cartilage-to-bone transition by ensuring cartilaginous template maturation during endochondral ossification, thus limiting premature bone formation. Collectively, our results indicate that BMP3 plays an important role in bone formation, exhibiting both inhibitory and regulatory properties in the context of ectopic bone formation, which may have important implications for therapeutic strategies in bone regeneration and skeletal diseases.

## Figures and Tables

**Figure 1 biomedicines-13-02508-f001:**
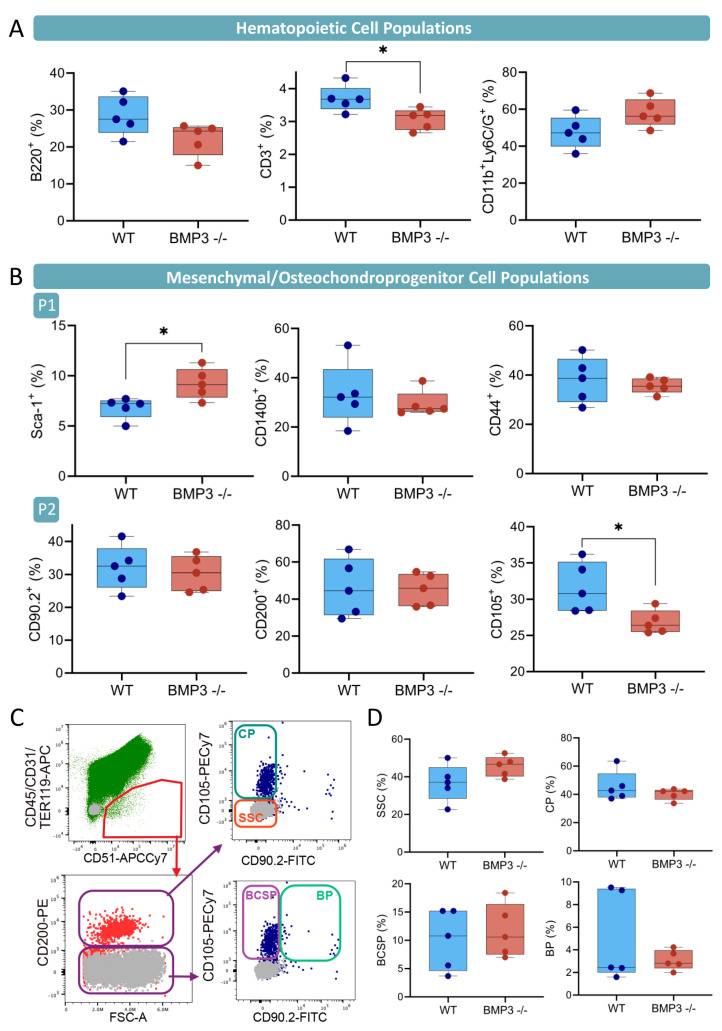
Cellular composition of bone marrow in WT and Bmp3^-/-^ mice assessed using flow cytometry. For flow cytometry analysis, single-cell suspensions were stained with hematopoietic lineage panel (anti-mouse B220-FITC, Ly6C/Ly6G-PECy7, CD11b-PE, CD3-APC, and F4/80-APCCy7), mesenchymal lineage panel 1 (anti-mouse CD90.2-FITC, CD200-PE, CD105-PECy7, CD45-APC, TER119-APC, CD31-APC, and CD51-biotin/streptavidin-APCeF780), and mesenchymal lineage panel 2 (anti-mouse Sca-1-FITC, CD140b-PE, CD45-APC, TER119-APC, CD31-APC, and CD44-APCCy7). Dead cells were excluded via binding of 7-AAD. (**A**) Proportions of hematopoietic lineage cells were determined among single, live cells. (**B**) Proportions of Sca-1^+^, CD140b^+^, and CD44^+^ cells were determined among CD45^−^TER119^−^CD31^−^ single, live cells. (**C**) Proportions of CD200^+^, CD105^+^, and CD90.2^+^ cells were determined between CD45^−^TER119^−^CD31^−^CD51^+^ single, live cells. (**D**) Proportions of earliest bone progenitors (SSC, CD200^+^CD90.2^−^CD105^−^) and cartilage progenitors (CP, CD200^+^CD105^+^) were determined amongst CD45^−^TER119^−^CD31^−^CD51^+^CD200^+^ single, live cells. Proportions of committed progenitors (BCSP, CD200^−^CD90.2^−^CD105^+^) and bone progenitor (BP, CD200^−^CD90.2^+^CD105^+^) cells were determined amongst CD45^−^TER119^−^CD31^−^CD51^+^CD200^−^ single, live cells. Dot plots represent delineation of subpopulations of CD200^+^ and CD200^−^ cells; grey dots, non-stained control cells; colored dots, labeled cells. Data are representative of 2 separate experiments. Horizontal lines and bars are medians and interquartile range (IRQ) (WT, n = 5; Bmp3^-/-^, n = 5). Statistical significance was assessed based on the Mann–Whitney test, * *p* < 0.05 values are marked on the plots with asterisks.

**Figure 2 biomedicines-13-02508-f002:**
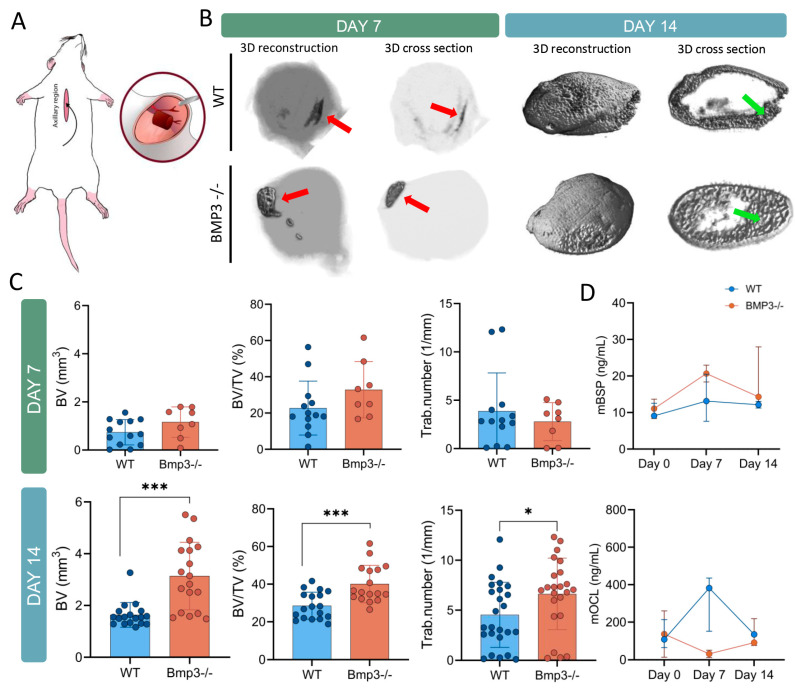
Assessment of ectopic bone formation based on micro-CT quantification. Graphic representation of implantation procedure within axillary region (**A**). 3D reconstruction and cross section of calcified implants sampled 7 and 14 days following implantation. Red arrows indicate the location of bone tissue within the whole implant on day 7, while green arrows indicate trabecular bone within the implant on day 14 (**B**). Micro CT quantification of bone parameters, bone volume (BV), bone volume fraction (BV/TV), and trabecular number (Tb.N.) in implants sampled 7 and 14 days after implantation. * *p* < 0.05, *** *p* < 0.001. (**C**). Bone remodeling marker values of bone sialoprotein (BSP) and osteocalcin (OCL) in serum 7 and 14 days post-implantation (**D**).

**Figure 3 biomedicines-13-02508-f003:**
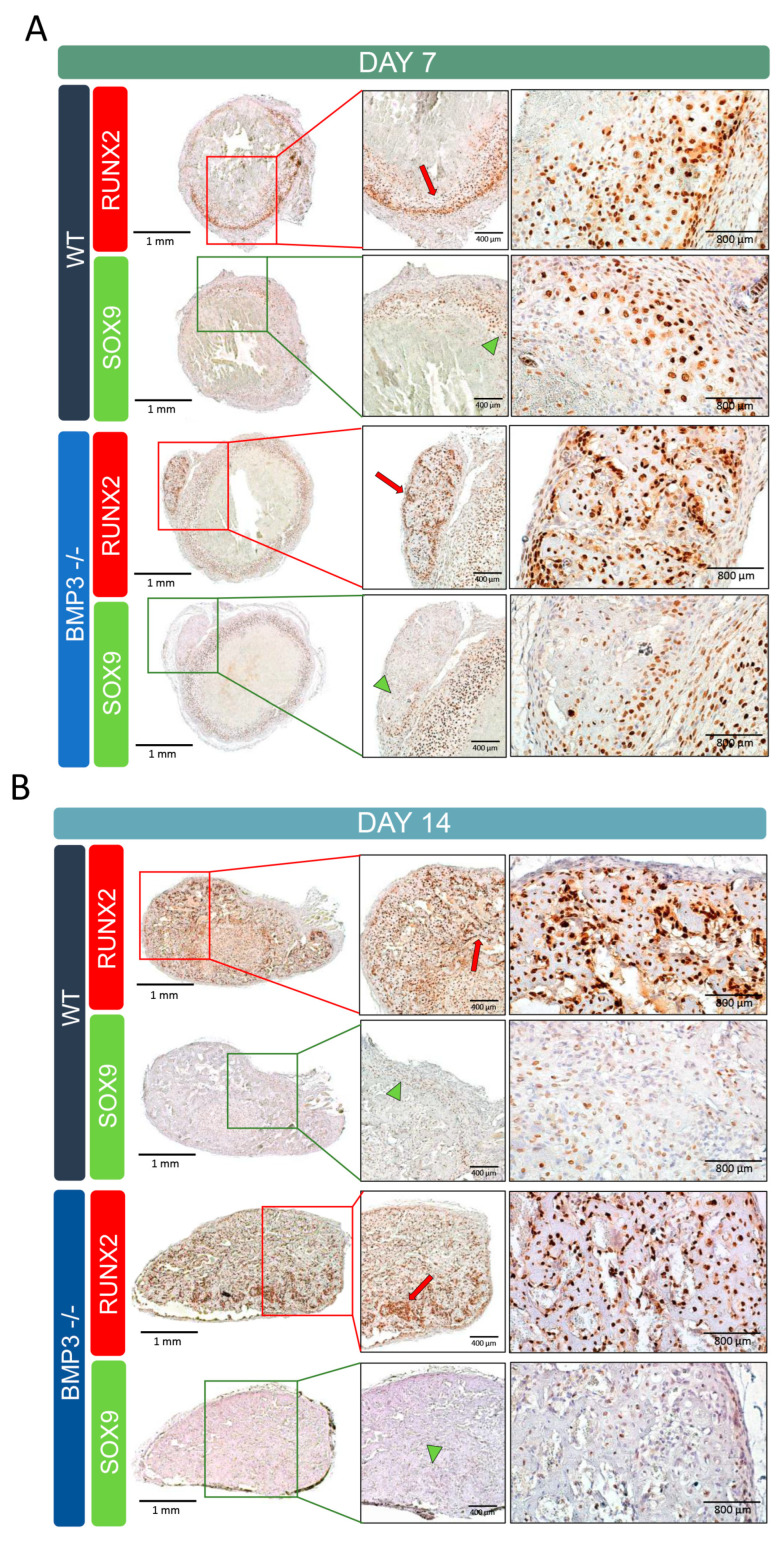
Immunohistochemical analysis of ectopic bone formation. Localization of Sox9 and Runx2 in WT and Bmp3^-/-^ mice 7 (**A**) and 14 (**B**) days after autologous blood coagulum with BMP6 implantation. Red arrows indicate Runx2 expression, while green arrowheads indicate Sox9 expression.

**Figure 4 biomedicines-13-02508-f004:**
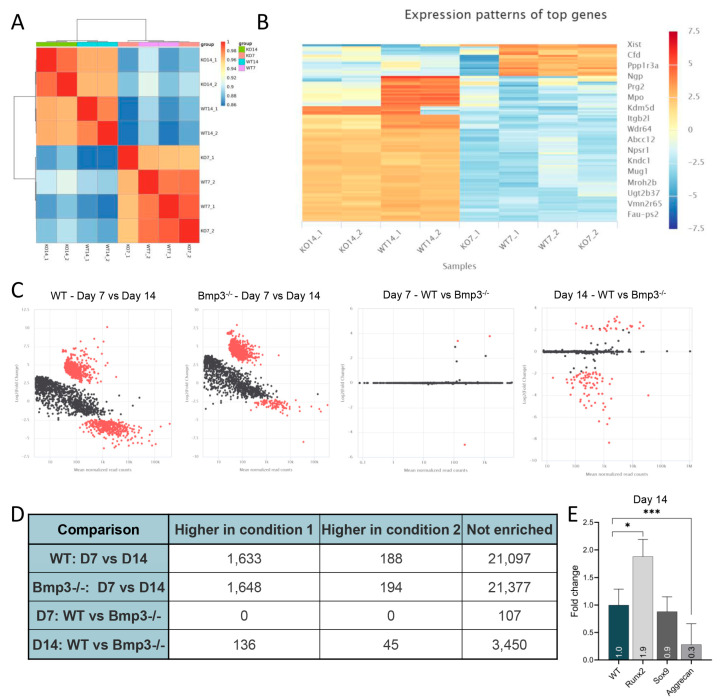
Transcriptomic analysis of ectopic bone in WT and Bmp3^-/-^ mice 7 and 14 days post-implantation. Sample similarity matrix (**A**). Expression patterns of top genes in each sample (**B**). Differential gene expression levels on day 7 and 14 for WT and Bmp3^-/-^ (**C**). Comparison of number of differentially enhanced genes between WT and Bmp3^-/-^ mice for day 7 and 14 (**D**). RT-qPCR for bone/cartilage marker genes in implants after 14 days, * *p* < 0.05 *** *p* < 0.001 (**E**).

**Table 1 biomedicines-13-02508-t001:** Primers used for qPCR.

Primer	5′-3′	3′-5′
*Gapdh*	AGGTCGGTGTGAACGGATTTG	TGTAGACCATGTAGTTGAGGTCA
*Runx2*	GAGGGACTATGGCGTCAAACA	GGATCCCAAAAGAAGCTTTGC
*Sox9*	CGTCAACGGCTCCAGCA	TGCGCCCACACCATGA
*Aggrecan*	CCTGCTACTTCATCGACCCC	AGATGCTGTTGACTCGAACCT

## Data Availability

The data presented in this study are available upon request made to the corresponding author. Due to ongoing research, the data has not yet been deposited into a trusted repository.

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
