# Peer review of "BMP3 Deficiency Accelerates Cartilage-to-Bone Transition in Ectopic Bone"

_biomedicines, 2025, doi:10.3390/biomedicines13102508_

Round 1
Reviewer 1 Report
Comments and Suggestions for Authors
The manuscript entitled “BMP3 Deficiency Accelerates Cartilage to Bone Transition in Ectopic Bone”, aims the effect of BMP3 on bone formation and remodeling, Bmp3-/- mice were subcutaneously implanted with autologous blood coagulum device containing BMP6 and bone formation.
Below are some suggestions:
- I suggest that the authors clarify the objective and also provide a better description of the methodology (e.g., which groups, number of animals, etc.).
- I suggest to the authors in the introduction:
- more information is needed; it would be interesting to divide it into topics, for example: molecular mechanisms of bone regeneration; role of BMPs....
- the introduction contains paragraphs with information on methodology (line 49).
- I suggest starting the methodology by creating a diagram (experimental design), including information about the groups, procedures performed, applications, etc.
- In Figure 2 (Results), it would be better to divide it into two plates (A/B) and (C/D). The authors should insert arrows and asterisks in the micro-CT images to indicate the changes observed. They should also include this in the caption.
- Authors should begin the discussion with a paragraph summarizing what was done, as well as the objective and main results.
- For a more solid scientific basis, I suggest including a greater number of references and updates (from the last five years).
Comments on the Quality of English Language
Moderate editing.
Author Response
The manuscript entitled “BMP3 Deficiency Accelerates Cartilage to Bone Transition in Ectopic Bone”, aims the effect of BMP3 on bone formation and remodeling, Bmp3-/- mice were subcutaneously implanted with autologous blood coagulum device containing BMP6 and bone formation.
Below are some suggestions:
1) I suggest that the authors clarify the objective and also provide a better description of the methodology (e.g., which groups, number of animals, etc.).
RESPONSE: We thank the reviewer for the insightful comments. We have added the information regarding the number of animals per group for each experiment for better interpretation of our results and statistical methodology.
- I suggest to the authors in the introduction:
2) more information is needed; it would be interesting to divide it into topics, for example: molecular mechanisms of bone regeneration; role of BMPs....
RESPONSE: Since the Journal does not dictate structured Introduction with Sections or Topics we tried to divide the Introduction into paragraphs corresponding to the specific topics ( bone tissue formation, ectopic/heterotopic bone formation and division, general BMP information and more information on BMP3, as a focus of our research, and brief summary of our experimental design. We have further described the regulation of bone tissue by BMPs in this section (Line 68-73).
3) the introduction contains paragraphs with information on methodology (line 49).
RESPONSE: Although there are no specific methodology information, we agree with the reviewer and have removed part of the sentence that imply on methodology (New version Lines 50-51).
4) I suggest starting the methodology by creating a diagram (experimental design), including information about the groups, procedures performed, applications, etc.
RESPONSE: We thank the reviewer for the comment. We have included a new figure, as a Supplement Figure 1. as a diagram of the experimental design of our work.
5) In Figure 2 (Results), it would be better to divide it into two plates (A/B) and (C/D). The authors should insert arrows and asterisks in the micro-CT images to indicate the changes observed. They should also include this in the caption.
RESPONSE: We thank the author for the comment and suggestion. We have noticed that the figure caption did not elaborate on the image in the most accurate manner. We have corrected the caption so it corresponds to the latest version of the image and added arrows to indicate the changes between Bmp3-/- and WT mice, and we also changed the color scheme for Day 7 and 14 denominations, not to introduce confusion with WT and Bmp3-/- mice.
6) Authors should begin the discussion with a paragraph summarizing what was done, as well as the objective and main results.
RESPONSE: As the reviewer commented, we have added a paragraph summarizing our aim and results that we further discussed in the rest of the Discussion section.
7) For a more solid scientific basis, I suggest including a greater number of references and updates (from the last five years).
RESPONSE: As the reviewer suggested, we have updated the Introduction and Discussion to further describe the problematics and interpretation of our results with referring to more recent papers published in the field of bone tissue and BMPs with references number 4. 5, 14, 15, 16, 43.
Reviewer 2 Report
Comments and Suggestions for Authors
The paper under review explores the role of BMP3 in regulating ectopic bone formation and the transition from cartilage to bone tissue, using a mouse knockout model combined with micro-CT, histology, flow cytometry, and transcriptomic profiling. The originality of the work lies in its systematic focus on BMP3, a member of the BMP family that is less studied compared to its osteogenic counterparts. While BMP3 has been known as an antagonist of osteogenic BMPs, this study advances understanding by linking its absence to accelerated ossification and altered bone marrow composition, thus positioning it as an important regulator in bone remodeling. This is a relevant and timely contribution, as it connects developmental signaling pathways with potential implications for regenerative medicine and skeletal disease.
The methodology is clearly explained and draws on well-established experimental models. The ectopic bone formation assay using a blood coagulum device supplemented with BMP6 is appropriate, as it minimizes confounding mechanical factors and allows emphasis on molecular and cellular regulation. The inclusion of flow cytometry to characterize marrow cell composition, along with RNA sequencing to capture transcriptomic changes, provides a multidimensional view of BMP3's role. Micro-CT and histological confirmation strengthen the conclusions by directly linking molecular observations to structural outcomes. The methodology therefore appears robust, though certain points could benefit from clarification, such as more detail on statistical approaches, sample size justification, and whether randomization or blinding was applied during histological evaluations.
In terms of repeatability, the work is largely replicable, as the experimental protocols for mouse models, FACS panels, RNAseq pipelines, and bone imaging are documented in sufficient detail, including reagents, instruments, and software versions. The bioinformatic analysis is explicitly described using reproducible workflows such as Nextflow, which enhances transparency. However, the small sample numbers in RNAseq (two per group) reduce reproducibility and statistical power at the transcriptomic level, making validation with RT-qPCR crucial, though more extensive replication would be preferable.
The results are consistent and well-illustrated, showing that BMP3 deficiency correlates with more advanced ossification, increased trabecular bone formation, and changes in osteochondroprogenitor subsets. The transcriptomic data support these findings by showing strong upregulation of osteogenic and myeloid-associated genes. The integration of molecular and histological evidence is convincing. Still, the interpretation occasionally appears somewhat overstated, particularly when ascribing regulatory roles to BMP3 in marrow-immune interactions without fully dissecting causality.
Regarding the mathematical or quantitative aspects, the paper does not rely heavily on equations but on statistical comparisons and fold-change reporting in gene expression. The formulas presented, such as the 2^(-ΔΔCT) method for qPCR analysis, are standard and correctly reported, with no evident errors. Reported statistical thresholds (*p < 0.05, ***p < 0.001) are conventional, but the lack of correction for multiple comparisons in RNAseq validation could inflate significance levels. Clarification on whether such corrections were applied would improve rigor.
The findings presented in the paper can be effectively implemented in mechanical models, as demonstrated by recent advancements [1,2], to enhance the predictive accuracy of these models. A discussion on this aspect would further strengthen the impact of the paper.
[1] Allena, R., Scerrato, D., Bersani, A. M., and Giorgio, I. (2025). A bi-dimensional model bridging microdamage evolution and bone remodeling: a computational study on a human femur. Mathematics and Mechanics of Complex Systems, 13(3), 347-376.
[2] Bednarczyk, E., and Lekszycki, T. (2022). Evolution of bone tissue based on angiogenesis as a crucial factor: new mathematical attempt. Mathematics and Mechanics of Solids, 27(6), 976-988.
In summary, the paper is original and well-structured, with strong methodological integration across imaging, molecular, and cellular levels. Its main strengths are the novelty of addressing BMP3's inhibitory role in ossification, the comprehensiveness of the methods, and the alignment of results across different experimental approaches. Its main weaknesses are the limited RNAseq replication, potential overinterpretation of causality in immune–bone crosstalk, and incomplete details on certain statistical aspects. Nonetheless, the study provides a meaningful contribution to bone biology and could have some value in understanding bone regeneration and disease.
Author Response
The paper under review explores the role of BMP3 in regulating ectopic bone formation and the transition from cartilage to bone tissue, using a mouse knockout model combined with micro-CT, histology, flow cytometry, and transcriptomic profiling. The originality of the work lies in its systematic focus on BMP3, a member of the BMP family that is less studied compared to its osteogenic counterparts. While BMP3 has been known as an antagonist of osteogenic BMPs, this study advances understanding by linking its absence to accelerated ossification and altered bone marrow composition, thus positioning it as an important regulator in bone remodeling. This is a relevant and timely contribution, as it connects developmental signaling pathways with potential implications for regenerative medicine and skeletal disease.
1) The methodology is clearly explained and draws on well-established experimental models. The ectopic bone formation assay using a blood coagulum device supplemented with BMP6 is appropriate, as it minimizes confounding mechanical factors and allows emphasis on molecular and cellular regulation. The inclusion of flow cytometry to characterize marrow cell composition, along with RNA sequencing to capture transcriptomic changes, provides a multidimensional view of BMP3's role. Micro-CT and histological confirmation strengthen the conclusions by directly linking molecular observations to structural outcomes. The methodology therefore appears robust, though certain points could benefit from clarification, such as more detail on statistical approaches, sample size justification, and whether randomization or blinding was applied during histological evaluations.
RESPONSE: We thank the reviewer for the comment. As noted we have added more details on statistical methodology and justification of the sample size. For histological sections we did not use randomization, but average samples per group that were quantified by micro-CT were selected.
2) In terms of repeatability, the work is largely replicable, as the experimental protocols for mouse models, FACS panels, RNAseq pipelines, and bone imaging are documented in sufficient detail, including reagents, instruments, and software versions. The bioinformatic analysis is explicitly described using reproducible workflows such as Nextflow, which enhances transparency. However, the small sample numbers in RNAseq (two per group) reduce reproducibility and statistical power at the transcriptomic level, making validation with RT-qPCR crucial, though more extensive replication would be preferable.
RESPONSE: As the reviewer noted, the sample size of 2, for RNAseq, seems low, but the sequencing depth of at least 30 millions per sample ensured that we get robust results. The sample preparation is also well established in our laboratory, ensuring quality RNA for library construction. We included a larger number of samples in qPCR confirmation, as this methodology requires more samples to obtain statistically significant results. We managed to confirm the results using two different approaches.
3) The results are consistent and well-illustrated, showing that BMP3 deficiency correlates with more advanced ossification, increased trabecular bone formation, and changes in osteochondroprogenitor subsets. The transcriptomic data support these findings by showing strong upregulation of osteogenic and myeloid-associated genes. The integration of molecular and histological evidence is convincing. Still, the interpretation occasionally appears somewhat overstated, particularly when ascribing regulatory roles to BMP3 in marrow-immune interactions without fully dissecting causality.
RESPONSE: As the reviewer noted, we reformulated and removed parts of the text where we over-interpreted causality in immune bone crosstalk and the potential role of BMP3 in these processes and further described statistical analyses used for data analysis.(Line 530-535 and 550-552).
4) Regarding the mathematical or quantitative aspects, the paper does not rely heavily on eq-uations but on statistical comparisons and fold-change reporting in gene expression. The formulas represented, such as the 2^(-ΔΔCT) method for qPCR analysis, are standard and correctly reported, with no evident errors. Reported statistical thresholds (*p < 0.05, ***p < 0.001) are conventional, but the lack of correction for multiple comparisons in RNAseq validation could inflate significance levels. Clarification on whether such corrections were applied would improve rigor.
RESPONSE: We thank the reviewer for the comment. Although the adjustment in the p-value was used to filter out falsely significant genes, it was not described in the text, so we added this description (New version Line 219-220). For the classical qPCR results, we used conventional p-values, as this methodology is well established and the number of samples was higher.
5) The findings presented in the paper can be effectively implemented in mechanical models, as demonstrated by recent advancements [1,2], to enhance the predictive accuracy of these models. A discussion on this aspect would further strengthen the impact of the paper.
[1] Allena, R., Scerrato, D., Bersani, A. M., and Giorgio, I. (2025). A bi-dimensional model bridging microdamage evolution and bone remodeling: a computational study on a human femur. Mathematics and Mechanics of Complex Systems, 13(3), 347-376.
[2] Bednarczyk, E., and Lekszycki, T. (2022). Evolution of bone tissue based on angiogenesis as a crucial factor: new mathematical attempt. Mathematics and Mechanics of Solids, 27(6), 976-988.
RESPONSE: We thank the reviewer for the suggestion. We have included the second reference in the Discussion part, as it adds to the interpretation of our results, but the first is not in the scope of this research (reference 43), as it focuses on the fracture repair, not on the new bone formation.
6) In summary, the paper is original and well-structured, with strong methodological integration across imaging, molecular, and cellular levels. Its main strengths are the novelty of addressing BMP3's inhibitory role in ossification, the comprehensiveness of the methods, and the alignment of results across different experimental approaches. Its main weaknesses are the limited RNAseq replication, potential overinterpretation of causality in immune–bone crosstalk, and incomplete details on certain statistical aspects. Nonetheless, the study provides a meaningful contribution to bone biology and could have some value in understanding bone regeneration and disease.
RESPONSE: As the reviewer noted, we reformulated and removed parts of the text where we over-interpreted causality in immune bone crosstalk and the potential role of BMP3 in these processes and further described statistical analyses used for data analysis.(Line 527-530 and 547-549).
Round 2
Reviewer 1 Report
Comments and Suggestions for Authors
I thank the authors for implementing the suggestions from the review process.
Comments on the Quality of English LanguageModerate editing.